# Using Genomics Feature Selection Method in Radiomics Pipeline Improves Prognostication Performance in Locally Advanced Esophageal Squamous Cell Carcinoma—A Pilot Study

**DOI:** 10.3390/cancers13092145

**Published:** 2021-04-29

**Authors:** Chen-Yi Xie, Yi-Huai Hu, Joshua Wing-Kei Ho, Lu-Jun Han, Hong Yang, Jing Wen, Ka-On Lam, Ian Yu-Hong Wong, Simon Ying-Kit Law, Keith Wan-Hang Chiu, Jian-Hua Fu, Varut Vardhanabhuti

**Affiliations:** 1Department of Diagnostic Radiology, Li Ka Shing Faculty of Medicine, University of Hong Kong, Hong Kong; chenyix@hku.hk (C.-Y.X.); kwhchiu@hku.hk (K.W.-H.C.); 2Department of Thoracic Surgery, Sun Yat-sen University Cancer Center, Guangzhou 510000, China; huyih@sysucc.org.cn (Y.-H.H.); yanghong@sysucc.org.cn (H.Y.); 3State Key Laboratory of Oncology in South China, Collaborative Innovation Center for Cancer Medicine, Guangzhou 510000, China; hanlj@sysucc.org.cn (L.-J.H.); wenjing@sysucc.org.cn (J.W.); 4Guangdong Esophageal Cancer Institute, Guangzhou 510000, China; 5School of Biomedical Sciences, Li Ka Shing Faculty of Medicine, University of Hong Kong, Hong Kong; jwkho@hku.hk; 6Department of Medical Imaging, Sun Yat-sen University Cancer Center, Guangzhou 510000, China; 7Department of Clinical Oncology, Li Ka Shing Faculty of Medicine, University of Hong Kong, Hong Kong; lamkaon@hku.hk; 8Department of Surgery, Li Ka Shing Faculty of Medicine, University of Hong Kong, Hong Kong; iyhwong@hku.hk (I.Y.-H.W.); slaw@hku.hk (S.Y.-K.L.)

**Keywords:** esophageal squamous cell carcinoma, neoadjuvant chemoradiotherapy, prognosis, radiogenomic

## Abstract

**Simple Summary:**

Prognosis for patients with locally advanced esophageal squamous cell carcinoma (ESCC) remains poor mainly due to late diagnosis and limited curative treatment options. Neoadjuvant chemoradiotherapy (nCRT) plus surgery is considered the standard of care for patients with locally advanced ESCC. Currently, predicting prognosis remains a challenging task. Quantitative imaging radiomics analysis has shown promising results, but these methods are traditionally data-intensive, requiring a large sample size, and are not necessarily based on the underlying biology. Feature selection based on genomics is proposed in this work, leveraging differentially expressed genes to reduce the number of radiomic features allowing for the creation of a CT-based radiomic model using the genomics-based feature selection method. The established radiomic signature was prognostic for patients’ long-term survival. The radiomic nomogram could provide a valuable prediction for individualized long-term survival.

**Abstract:**

Purpose: To evaluate the prognostic value of baseline and restaging CT-based radiomics with features associated with gene expression in esophageal squamous cell carcinoma (ESCC) patients receiving neoadjuvant chemoradiation (nCRT) plus surgery. Methods: We enrolled 106 ESCC patients receiving nCRT from two institutions. Gene expression profiles of 28 patients in the training set were used to detect differentially expressed (DE) genes between patients with and without relapse. Radiomic features that were correlated to DE genes were selected, followed by additional machine learning selection. A radiomic nomogram for disease-free survival (DFS) prediction incorporating the radiomic signature and prognostic clinical characteristics was established for DFS estimation and validated. Results: The radiomic signature with DE genes feature selection achieved better performance for DFS prediction than without. The nomogram incorporating the radiomic signature and lymph nodal status significantly stratified patients into high and low-risk groups for DFS (*p* < 0.001). The areas under the curve (AUCs) for predicting 5-year DFS were 0.912 in the training set, 0.852 in the internal test set, 0.769 in the external test set. Conclusions: Genomics association was useful for radiomic feature selection. The established radiomic signature was prognostic for DFS. The radiomic nomogram could provide a valuable prediction for individualized long-term survival.

## 1. Introduction

Esophageal cancer (EC) accounted for 572,034 new cases and 508,585 deaths of cancer overall worldwide in 2018, ranking seventh in incidence and sixth terms of mortality [1]. There is geographic variation in EC pathological subtype incidence. Approximately 90% of EC cases at the time of diagnosis in China are esophageal squamous cell carcinoma (ESCC) [2]. Because of late-stage cancer diagnosis and limited clinical curative modality, the five-year overall survival (OS) rates for ESCC patients range from 15% to 25% [3]. Surgery has a central role in disease management, but a large proportion of patients showed local or distant metastasis after surgery within 3 years [4,5]. The addition of adjuvant chemotherapy or radiation therapy has proven to improve survival in patients with involved lymph node disease [6,7]. However, a recent network meta-analysis showed that adjuvant chemotherapy or radiation therapy could not significantly reduce death risk compared with surgery alone [8]. As shown by well-powered prospective randomized clinical trials, neoadjuvant chemoradiotherapy (nCRT) could benefit patients by improving tumor resection rate and long-term survival for patients with locally advanced EC [9,10,11,12]. The CROSS study demonstrated that patients receiving nCRT followed by surgery have a significantly increased median overall survival than those receiving surgery alone (49.4 vs. 29.0 months, *p* = 0.003) [9]. The Chinese study NEOCRTEC5010 also showed that compared with a simple surgery, patients’ median survival was improved from 66.5 to 100.1 months (*p* = 0.025) [10]. However, due to tumor heterogeneity, not all patients could gain a survival benefit from nCRT treatment. Currently, predicting prognosis remains a challenging task for ESCC patients. The ability to identify patients with poor prognoses is required for more effective personalized disease management.

Computed tomography (CT) is a broadly used non-invasive tool for disease assessment. For the evaluation of tumor malignancy and patient prognosis, the visualization of tumor heterogeneity is of vital importance. Radiomics are now widely used in prognostic prediction tasks for many tumor types [13,14,15]. Previous quantitative imaging research has generally focused on the prediction of the nCRT treatment effect in EC [16,17,18,19]. Furthermore, the majority of esophageal cancer studies were based on imaging data acquired from a single time point [20,21]. Delta radiomics has been proposed recently that reflected changes of radiomic features across certain therapies. Delta radiomic features were reported to improve model performance for different tasks of a variety of cancer, including diagnosis [22], therapy response evaluation [23,24,25,26], and prognosis prediction [27,28].

Methods to combat the “curse of high dimensionality” intrinsic to radiomics methodology are important for the construction of radiomics prediction models. Methods that can adequately perform feature reduction will enable a more accurate predictive performance and decreased computational costs. Widely used methods mainly focus on data-driven feature selection. For the development of novel models, Wynants et al. [29] recommended the application of prior knowledge and expert opinion for the selection of significant features, rather than choosing a predictor using an entirely data-driven method [30]. This could be more important for some specific clinical scenarios with a small sample size [31].

Previous studies have shown that image-derived radiomic features are the reflection of the underlying molecular changes in the tumor cells with phenotypic consequences. Segal et al. demonstrated that semantic image traits correlated with genetic profiles in human hepatocellular carcinomas (HCC) patients [32]. Kuo and his colleagues showed that genomics analysis helps the identification of patterns of gene expression related to drug response in HCC [33]. Grossmann et al. [34] identified and independently validated 13 radiomic-pathway modules with coherent expression patterns. Eleven of the modules were significantly associated with overall survival, staging, or histology. Panth et al. [35] further proved a causal relationship between gene expression and imaging traits. Traditionally, researchers have concentrated efforts to reveal the association between radiomics and genomic data (including transcriptomic data). To our knowledge, no prior studies have investigated the feasibility of genomics-driven radiomics feature selection.

We hypothesize that genomics-driven feature selection of radiomic features will lead to a more robust and generalizable predictive model. The objective of this study is to perform a proof-of-concept study using genomics data as a method for feature selection of radiomics applied to CT scans to demonstrate value when constructing a radiomics-based prognostic prediction model for ESCC patients receiving nCRT followed by surgery.

## 2. Materials and Methods

### 2.1. Patients Cohorts

The experimental design of this research was depicted in Figure 1. Patient records from April 2007 to December 2016 were retrieved from the Sun Yat-sen University Cancer Center, Guangzhou, China (institution 1) and the University of Hong Kong, Hong Kong (institution 2). The selection criteria included: (a) patients aged 18–80 years; (b) had histologically confirmed ESCC; (c) had standardized baseline and post-nCRT enhanced CT scans; and (d) received nCRT plus surgery. The exclusion criteria included: (a) patients who underwent anticancer treatments before the baseline CT scans; (b) with a history of other malignancies; and (c) with incomplete medical records. For institution 1, patients received 75 mg/m^2^ cisplatin on day 1 and 25 mg/m^2^ vinorelbine on days 1 and 8 for 2 cycles, or 25 mg/m^2^ cisplatin and 25 mg/m^2^ docetaxel on days 1, 8, 15, and 21, with a total dose of 40 or 44 Gy for concurrent radiotherapy. For institution 2, 50 mg/m^2^ paclitaxel and carboplatin AUC 2 for 5 cycles or 100 mg/m^2^ cisplatin and 500 mg/m^2^ fluorouracil were administrated intravenously for 4 days for weeks 1 and 5, with a total dose of 40 or 41.4 Gy for concurrent radiotherapy. Patients with biopsy samples with genetic profiles from institution 1 were allocated to the training group. The rest from institution 1 was used as the internal test group. Patients from institution 2 were allocated to the external test group. This study has been approved by institutional review boards from both institutions (refer to Appendix A for further detail). Patients from institution 1 were part of a prior prospective study [10]. Due to the retrospective nature, informed consent from patients was waived.

### 2.2. Data Extraction and Feature Selection

#### 2.2.1. Radiomic Features Extraction and Preprocessing

Regions of interests (ROIs) were manually segmented using ITK-SNAP, and radiomic features extraction was performed by the open-source Python package PyRadiomics [36]. Both original and wavelet-filtered features were extracted. There were three main groups of radiomic features: (1) pre-nCRT features; (2) post-nCRT features; (3) delta features (Δfeatures). The Δfeatures were the relative changes (expressed in percentage) of the radiomic features before and after nCRT treatment. To assess feature robustness, we conducted a test-retest study. Two radiologists (V.V. 10 years and L.H. 9 years’ experience) individually contoured the ROIs in the training set. Two groups of features were extracted, and those with intra-class correlation coefficients (ICC) > 0.80 were selected. To minimize the institutional difference, we used the ComBat method for feature harmonization [37]. The ComBat method has been commonly used in genomic studies and shown to successfully correct the multicenter differences in imaging features values resulting from different image acquisition protocols [38].

#### 2.2.2. Genomic Data

For twenty-eight ESCC patients from the training set, the pre-nCRT tissue samples were extracted from the primary tumor sites around 2 weeks before nCRT. The genomic profiles were measured using a GeneChip^®^ Human Genome U133 Plus 2.0 Array (Affymetrix, Santa Clara, CA, USA).

#### 2.2.3. Clinical Data and Follow-up

The primary endpoint was disease-free survival (DFS). DFS referred to the length of time from surgery to recurrence of tumor or death. OS, as the secondary endpoint, was the duration from recruitment to death or the last follow-up. The minimum follow-up period was 36 months after surgery. For patients with no documented clinical endpoints, their survival was censored on 31 December 2019.

The clinical records were reviewed by the surgeons and oncologists in charge. Clinical staging was evaluated according to the American Joint Committee on Cancer (AJCC) TNM staging system, 8th edition [38]. Patients’ clinical characteristics, including age, sex, treatment response, tumor length, tumor location, histologic grade, ΔBMI, smoking status, drinking status, and family history of tumor, were recorded.

### 2.3. Data Resampling

Given the imbalanced distribution of DFS (7 out 21 having recurrence), the synthetic minority oversampling technique has been adopted for data re-balancing in the training set [39].

### 2.4. Feature Selection Using Genomic Data

We introduced a feature selection step according to the features’ correlation with differentially expressed (DE) genes in the tumor biopsy sample. For twenty-eight ESCC patients from the training set, the “limma” package was applied to detect DE genes between patients with different survival outcomes. DE genes were ranked by the *p*-value according to the limma test, and the ordered list of DE genes was further analyzed by enrichment analysis of gene terms with the use of g: Profiler [40]. Data sources include gene ontology biological process and pathway collections from Kyoto Encyclopedia of Genes and Genomes, Reactome, and WikiPathways. The enrichment analysis found significant terms consisting of sets of genes. The overlapped genes were defined as DE genes that could be found in both significant gene sets and DE genes. The overlapped genes were used as a filter for the selection of radiomic features. Radiomic features that were significantly correlated with these overlapped genes using Pearson’s correlation test. Correlated radiomic features were included for further analysis.

### 2.5. Feature Selection Using Data-Driven Approach

In the training set, radiomic features were selected in three steps. First, correlated features (Pearson’s correlation coefficient larger than 0.80) were grouped, and each group of features was fitted into a DFS prediction model using a decision tree classifier. Features with the highest importance attributed by the model among features were considered the most important and retained. Second, according to univariate analysis, the top 100 best features predictive of DFS calculated were selected. Finally, we used regularized multivariate logistic regression with the least absolute shrinkage and selection operator (LASSO) penalty to further reduce the feature number [41]. The optimal λ was used to find predictors with non-zero coefficients.

### 2.6. Classification Model and Nomogram Construction

A linear regression model with selected features was built for the calculation of the radiomic score (Rad-score) after feature selection. The features selected from genomics feature selection and data-driven machine learning approaches were used to build classification model 1. The features selected from only data-driven approaches were used to build classification model 2.

We further built clinical nomograms integrating the radiological score and valuable clinical risk factors for prognosis prediction. Based on Cox proportional hazards model, hazard ratios (HRs) for Rad-score and other clinical variables were calculated. The correlations of the factors with DFS were further investigated in multivariable analyses. The final model was decided by the Akaike information criterion (AIC) in a backward selection manner. The selected risk factors were included to build clinical nomograms. Nomogram 1 and nomogram 2 were based on classification model 1 and classification model 2, respectively. The cut-off points for the nomograms were determined by Youden Index, to divide patients into different risk groups.

### 2.7. Statistical Analysis

Statistical analyses and graphic production were conducted using Python 3.7. and R 3.3.1. The “limma” method was used with a false discovery rate  <  0.05 and a two-fold difference as cut-off criteria. The enrichment analysis of gene terms of DE genes was conducted by g: Profiler with 0.05 as the cut-off for the false discovery rate. The correlation between radiomic features and overlapped genes were analyzed by Pearson correlation test and deemed significant if *p* < 0.05. We determined the prediction performance by area under the curve (AUC) of the receiver operating characteristic curve. Time-dependent AUCs for the multivariate Cox model performance were evaluated every 6 months. The calibration performance of the nomograms was measured graphically by calibration plots. Discrimination ability was tested by Harrell’s concordance index (C-index). We further employed decision curve analysis (DCA) to detect the net benefit brought by the nomograms for clinical settings [42]. The DFS and OS for patients stratified by nomogram predictions were shown by Kaplan–Meier curves and the statistical difference was measured using log-rank significance tests. Details could be found in the Appendix A.

## 3. Results

### 3.1. Patient Characteristics

Basic patient characteristics were listed in Table 1. A total of 106 ESCC patients receiving nCRT plus surgery treatment with follow-up information (mean age [standard deviation (SD)]: 59.01 [9.38]; male: 81.1%) were collected, including 65 from institution 1 and 41 from institution 2. Twenty-eight patients having biopsy samples with genetic profiles from institution 1 were used as the training set, the rest 37 were used as an internal test set, and 41 from institution 2 were allocated to the external test set (Appendix A). Apart from drinking status, the baseline characteristics were balanced between the training and the internal test set. There were differences in age, tumor location, clinical stage, and drinking status across the two institutions.

All patients had a minimum follow-up time of three years. The median follow-up period was 65.7 months (interquartile range (IQR), 20.4–88.6) for DFS and 70.1 (IQR, 31.3 –89.6) months for OS. The median follow-up period of DFS was 90.0 months (IQR, 60.8–98.6) for the training set, 74.4 months (IQR, 58.7–86.5) for the internal test set, and 37.7 months (IQR, 12.0–67.9) for the external test set.

### 3.2. Extracted Radiomic Features

We extracted 2553 radiomic features, including 851 features from each of the pre-nCRT CT, post-nCRT CT, and delta categories (107 original and 744 with wavelet filtration). For the feature robustness test, 2336 features with ICC > 0.80 were included for feature selection.

### 3.3. Feature Selection Using Genomic Data and Using Data-Driven Machine Learning Approaches

The statistical analysis in the training set found that 37 genes were differentially expressed between patients with and without relapse groups (*p* < 0.05) based on the limma test. The pathway enrichment analysis was conducted by using the ordered gene list generated from DE genes. Using these 37 DE genes, we identified 181 pathways that were enriched. Sixteen of the genes encompassed in enriched pathways overlap with 37 DE genes. The 16 overlapping genes were used for further correlation analysis (see Appendix A). After genomics feature selection, 35.4% (829 out of 2336) of radiomic features were found to be significantly correlated with at least one of the overlapped genes.

Radiomic features with and without genomics selection were both analyzed by the following machine learning process. After correlated feature elimination, univariate analysis, and LASSO selection, the feature number was finally reduced to eight. Radiomic features derived with and without genomics feature selection were listed in Appendix A. The correlation between selected features and DE genes was shown in Appendix A. Classification nomogram 1 consisted of eight features, all correlating with gene information. For classification nomogram 2, half features were not correlated with any discovered DE genes. Two features were selected for both nomograms.

### 3.4. Radiomic Classification Model

The radiomic classification model 1 consisted of radiomic features with genomics feature selection, which resulted in better performance and generalisability (AUC: 0.912 in the training set, 0.825 in the internal test set, 0.749 in the external test set), as shown in classification receiver operating characteristic curves (Figure 2a). The classification model 2 was constructed using radiomic features without genomics feature selection (AUC: 0.925 in the training set, 0.782 in the internal test set, and 0.679 in the external test set), as shown in Figure 2b. Prediction probabilities were used as Rad-score for further analysis.

### 3.5. Radiomics-Based Nomogram Construction

We proceeded to develop radiomic nomograms combining the Rad-score and clinicopathological factors. Clinical N staging was significantly correlated with DFS according to the univariate analysis. In the multivariate analysis with AIC stepwise selection for the construction of nomogram 1, Rad-score 1 (HR: 2.55; 95% CI: 1.85,3.52; *p* < 0.001), and clinical N staging (HR: 3.23; 95% CI: 1.66, 6.29; *p* < 0.001) were identified as independent prognostic factors in the Cox proportional hazards model and were incorporated into the nomogram 1 (Figure 3). Rad-score 2 (HR: 3.25; 95% CI: 1.98, 5.33; *p* < 0.001), and clinical N staging (HR: 2.76; 95% CI: 1.41, 5.38; *p* = 0.003) were also selected for nomogram 2.

The C-indexes of nomogram 1 and nomogram 2 were 0.869 and 0.875 in the training group, 0.812 and 0.757 in the internal test group, and 0.719 and 0.668 in the external test group, respectively. Time-dependent AUCs for the multivariate Cox model were evaluated (Figure 4). The performance of nomogram 1 and nomogram 2 for predicting 5-year DFS was assessed with respective AUCs of 0.912 and 0.918 in the training group, 0.852 and 0.810 in the internal test group, 0.769 and 0.724 in the external test group. The Cox model of the continuous radiomic signature also demonstrated good calibration for both nomograms (Appendix A). DCA confirmed the clinical benefits (Appendix A).

According to the optimal cut-off value (Youden index) for the diagnostic possibility of the clinical nomogram, the patients were classified as the low-risk group with diagnostic possibilities ≤ 0.51 and as the high-risk group with diagnostic possibilities > 0.51. Kaplan–Meier curves demonstrated that the risk stratification of nomogram 1 was associated with the DFS in the training group (*p* = 0.002) and internal test group (*p* < 0.001), and this finding was confirmed in the external test group (*p* < 0.001) (Figure 5). For nomogram 2, Kaplan–Meier curves showed good predictive value in the training group (*p* < 0.001) but less predictive in the internal test group (*p* = 0.032) and were not statistically different in the external test group (*p* = 0.220). Kaplan–Meier curves also showed better risk stratification of patients’ OS in nomogram 1 than that in nomogram 2 (Appendix A).

### 3.6. Prediction of Survival Status Using Delta Features

We further compared the changes of tumor volume in pre- and post-nCRT scans, and its correlation to patients’ survival was low (*p* = 0.571 for DFS, *p* = 0.215 for OS in KM curve analysis, Appendix A). The delta radiomic feature selected for both nomograms (Wavelet LLL filtered correlation from GLCM families) was significantly predictive of patients’ survival (*p* = 0.010 for DFS, *p* = 0.036 for OS in KM curve analysis). This GLCM radiomic feature was not associated with tumor volume (*p* = 0.482, Kruskal–Wallis test).

## 4. Discussion

Clinical risk stratification of patients is important for decision-making for exploring personalized treatment and for the prediction of prognostic outcome. The use of CT modality is widely available. Image assessment using radiomics is an emerging approach to predict response in cancer treatment and long-term survival. This is the first pilot study to include a two-time-point delta radiomics analysis in a prognostic prediction model in conjunction with a biological underpinning feature selection method. The constructed radiomics-based survival prediction in this study achieved good prognostic value with generalizability to the external test set, which could improve personalized management of ESCC patients, potentially improving clinical outcomes. For patients with ESCC treated by nCRT, Laruea et al. [43] developed a CT-based radiomic model with an AUC of 0.61 and borderline significant Kaplan–Meier curve result in the validation dataset (*p* = 0.053). Foley et al. [20] validated a PET-based prognostic model combined with clinical features in an international cohort, but this model did not enable significant discrimination between patient risk groups. Chen et al. [18] reported that a PET image feature model was independently associated with DFS and OS but with a small sample size (*n* = 32). Qiu et al. [44] constructed a nomogram using radiomic and clinical features based on one center data at one single time point. Our study established a radiomics-based clinical nomogram with a relatively larger sample size and validated on an external dataset. This was similar to previous findings that nomogram incorporating clinical factors and imaging features were of predictive value for EC patients [45,46,47]. Zhang et al. [45] reported that a nomogram based on clinical variables and imaging radiomic features was predictive of lymph node metastases. We proposed a potentially novel way of screening prognostic imaging texture features by gene-driven method.

With the advance of next-generation sequencing techniques and machine learning algorithms, there are increasing high-throughput omics data. While genomics and radiomics have been studied individually, the integration of genomic and radiomic data into multi-omics-based machine learning models could provide new scope for precision oncology, which would aid a more comprehensive understanding and management of cancer diseases.

A model incorporating delta imaging features may potentially capture characteristics of tumors’ response to treatment and therefore improve differentiation of tumor heterogeneity. Shrinkage of tumor volume was an important independent prognostic factor in EC patients, as it could improve the R0 resection rate [48,49]. Size-based measurement was commonly used in previous studies [16,50]. However, some cases may involve tumor necrosis, liquefaction, and fibrosis during the treatment process, without a significant decrease in tumor size. More recently, delta radiomic features from pre- and post-treatment were reported to improve cancer treatment response prediction, including chemoradiation therapy [24,25,26] and, more recently, immune therapy [51]. One delta radiomic feature (correlation from GLCM families) was significantly predictive of patients’ survival than the changes in tumor volume in our study. This is in accordance with previous studies that changes in radiomic features could outperform the volume measures for disease evaluation [23,24,46,52]. Tumor size-based evaluation could not account for some important factors such as spatial heterogeneity of primary tumor lesions that correlated with tumor biology. We demonstrated that radiomic features could have the potential to offer special insight to the tumor characterization as they could capture the advanced tumor heterogeneity that was not visible to the naked eyes. CT-based disease assessment could serve as effective tools for patients’ risk stratification.

Our results showed that genomics information was useful for radiomic feature selection. The nomogram constructed from radiomic features with genomics feature selection improved prediction value compared with the nomogram without genomics feature selection. Many genes have been proposed as prognostic predictors of ESCC [53]. Instead of analyzing the underlying biological process by gene enrichment analysis [54,55,56], we used these genomics data as a new feature selection filter to assure biological robustness. Radiomic features were correlated with overlapped genes, with the introduction of pathway information from enrichment analysis. Such lists of genes provided an additional biological correlation that is more than a simple correlation with the DFS outcome.

Machine learning feature selection methods are known to be prone to data-driven biases related to missing data, sample size and measurement errors. Machine learning selected features were recommended to make clinical sense from practitioners, as algorithms may underestimate clinically meaningful information [57]. The decreased predictive performance from internal to external test set could be largely correlated with overfitting in the training process and dissimilarities among data. The nomogram constructed from radiomic features with genomics feature selection showed better generalizable prediction. We provided a novel feature selection approach based on biological knowledge using genomics data to filter significant radiomic features. This method could achieve clinically important improvements and decrease indirect prejudices associated with data-driven algorithm estimation.

Pathological complete response (pCR) was reported to be a significant predictor for improved survival in EC patients [58]. In our analysis, the reason why pCR does not correlate with survival was not clear. One possible reason was the small sample size and sampling bias. The other reason may be binary division nCRT response to pCR and non-pCR may not fully reflect the tumor’s response to nCRT treatment, which may be of more graded response. More detailed evaluation criteria such as four categories of complete responses, partial responses, stable diseases, and progressive diseases [59] may provide a more comprehensive and reliable measure of response with increased predictive value.

As a pilot investigation, our study was limited by several aspects. The main limitation was the small sample size as data were retrospectively collected from prospective clinical studies to ensure the complete acquisition of images across nCRT, the accuracy of follow-up information, and accessibility of genetic profiles. This could lead to selection bias, and the presented clinical characteristics become decreasingly representative of the entire population. Secondly, due to the retrospective nature, part of the clinical variables was not balanced for different institutions, which is a common problem for cross-regional research [20]. Further prospective studies with larger sample sizes are required for further validation of the biological correlation between imaging features and genetic markers. Lastly, we used the correlation between radiological features and genetic data for patients with different disease recurrence outcome for feature selection. The gene-based correlation-filtered selection was chosen to avoid ambiguity in this pilot study. Future investigations on the pathway-guided selection process and the underlying driving biology are suggested.

## 5. Conclusions

Genomics association was useful as a method for radiomic feature selection. The established radiomic signature was prognostic for patients’ long-term survival. The radiomic nomogram could provide a valuable prediction for individualized long-term survival.

## Figures and Tables

**Figure 1 cancers-13-02145-f001:**
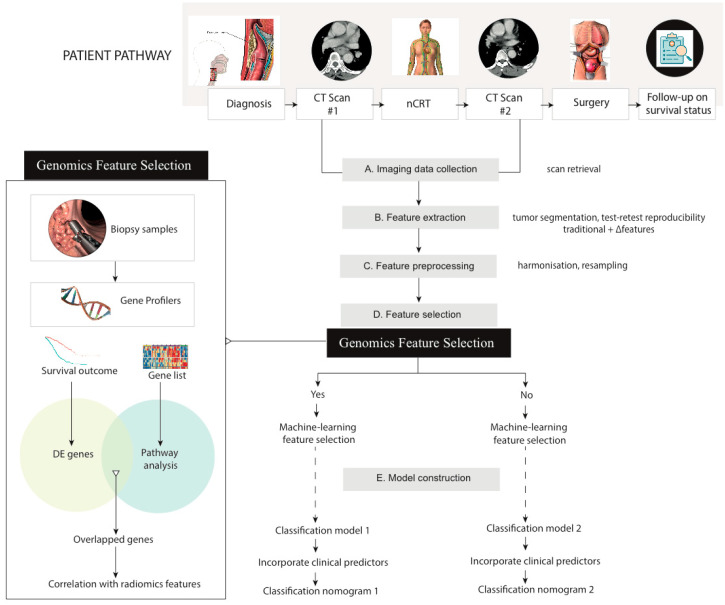
Study workflow. nCRT: neoadjuvant chemoradiotherapy; DE genes: differentially expressed genes.

**Figure 2 cancers-13-02145-f002:**
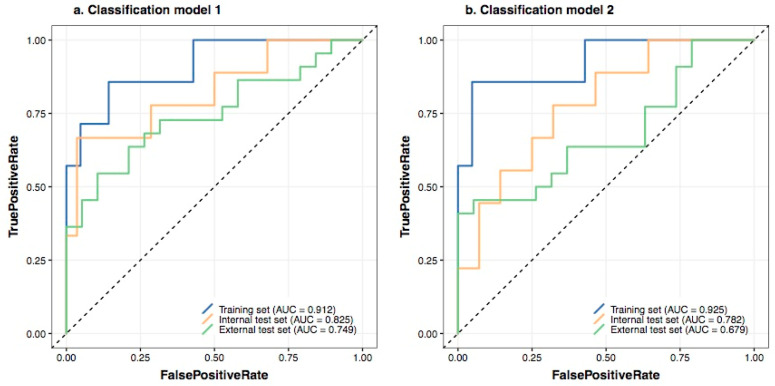
Prediction performance of radiomic classification models in the training set (blue color), internal test set (yellow color), and external test set (green color). (**a**) Classification model 1 with genomics feature selection. (**b**) Classification model 2 without genomics feature selection.

**Figure 3 cancers-13-02145-f003:**
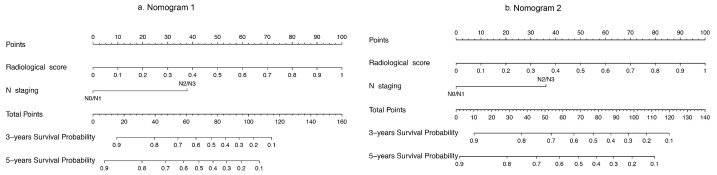
Prognostic nomograms. Probability of 3-year and 5-year disease-free survival of the nomograms developed by Rad-score and nodal staging information. (**a**) Nomogram 1 (with genomics feature selection). (**b**) Nomogram 2 (without genomics feature selection).

**Figure 4 cancers-13-02145-f004:**
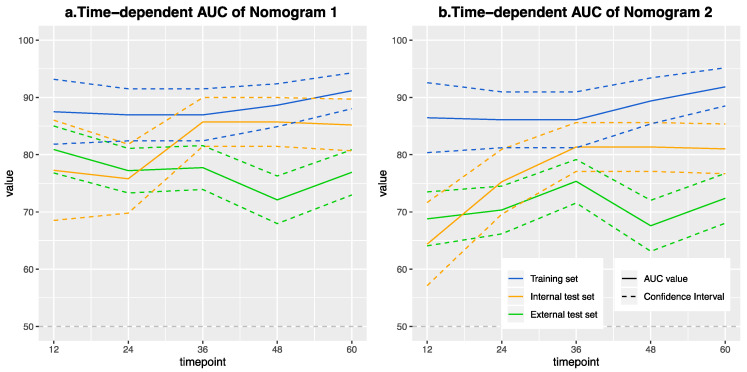
Time-dependent ROC curves for the nomograms in the training set (blue color), internal test set (yellow color), and external test set (green color). ROC, receiver operating characteristic. (**a**) Nomogram 1 with genomics feature selection. (**b**) Nomogram 2 without genomics feature selection. Dotted lines represent the 95% confidence interval (CI).

**Figure 5 cancers-13-02145-f005:**
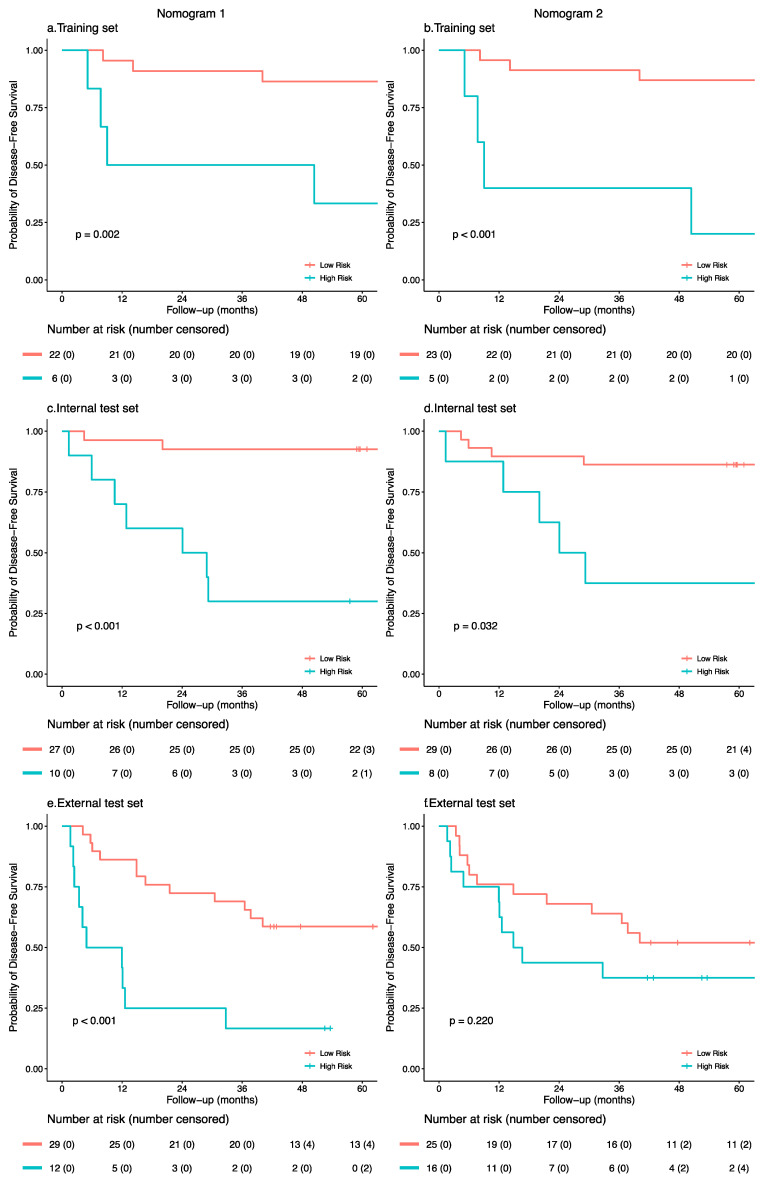
Disease-free survival for patients from high-risk (blue) and low-risk (pink) groups stratified by nomogram predictions. (**a**,**c**,**e**) Nomogram 1 with genomics feature selection. (**b**,**d**,**f**) Nomogram 2 without genomics feature selection. Kaplan–Meier curves showing disease-free survival in patients stratified by nomogram predictions in the training, internal test and external test set. Risk tables under Kaplan–Meier curves showing the number of patients at risk at 0, 12, 24, 36, 48, and 60 months. The difference between the two curves was compared by the log-rank test.

**Table 1 cancers-13-02145-t001:** Patients’ clinical characteristics.

Characteristic	Institution 1	Institution 2	*p*-Value
	65	41	
pCR (%)			1.00
No	34 (52.3)	22 (53.7)	
Yes	31 (47.7)	19 (46.3)	
Sex (%)			1.00
Male	53 (81.5)	33 (80.5)	
Female	12 (18.5)	8 (19.5)	
Age			<0.01
Mean (SD)	55.77 (6.79)	64.15 (10.64)	
cT staging 8th edition (%)			<0.01
1	1 (1.5)	0 (0.0)	
2	20 (30.8)	1 (2.4)	
3	44 (67.7)	39 (95.1)	
4	0 (0.0)	1 (2.4)	
cN staging 8th edition (%)			<0.01
0	7 (10.8)	1 (2.4)	
1	41 (63.1)	13 (31.7)	
2	17 (26.2)	23 (56.1)	
3	0 (0.0)	4 (9.8)	
Tumor location (%)			0.03
Proximal third	5 (7.7)	2 (4.9)	
Middle third	39 (60.0)	15 (36.6)	
Distal third	21 (32.3)	24 (58.5)	
Tumor Length (cm)			0.34
Mean (SD)	5.29 (1.99)	5.66 (1.87)	
Histologic grade (%)			0.85
1	5 (7.7)	3 (7.3)	
2	41 (63.1)	28 (68.3)	
3	19 (29.2)	10 (24.4)	
ΔBMI			0.81
Mean (SD)	0.01 (0.06)	0.02 (0.06)	
Tobacco use (%)			0.36
No	26 (40.0)	12 (29.3)	
Yes	39 (60.0)	29 (70.7)	
Drinking (%)			<0.01
No	49 (75.4)	18 (43.9)	
Yes	16 (24.6)	23 (56.1)	
Family tumor history (%)			0.55
No	55 (84.6)	32 (78.0)	
Yes	10 (15.4)	9 (22.0)	
DFS follow-up time (months), median [IQR]	77.82 [57.50, 93.63]	37.71 [11.97, 67.86]	<0.01
OS follow-up time (months), median [IQR]	79.36 [59.34, 96.30]	42.08 [19.17, 72.53]	<0.01

pCR: pathologic complete response; ΔBMI: change of body mass index from pretreatment to post-chemoradiation; DFS: disease-free survival; OS: overall survival; SD: standard deviation; IQR: interquartile range.

## Data Availability

The data that support the findings of this study are openly accessible in the NCBI Gene Expression Omnibus at www.ncbi.nlm.nih.gov/geo/ (GSE45670).

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
