# Peer review of "Using Genomics Feature Selection Method in Radiomics Pipeline Improves Prognostication Performance in Locally Advanced Esophageal Squamous Cell Carcinoma—A Pilot Study"

_cancers, 2021, doi:10.3390/cancers13092145_

Round 1
Reviewer 1 Report
Your paper is well written with clinically important message related to prognostic value of genomics-driven radiomics feature selection. However, there was no details or explanations about the methods: Did you use contrast CT? What are feature extractions or texture analysis from CT? How did you combine or overlap DE genes on CT features? CT is basically morphologic examination, and contrast enhancement is purely blood flow phenomenon. FDG PET demonstates functional or metabolical status of the cancer. Therefore, radiogenomics using PET/CT data seems better than only CT data.
Page 3, line 96: This could more.....------->This could be more....
Reviewer 2 Report
Overall summary:
Response prediction is an aspect which has been frequently discussed over the past years. Radiomics has also been recently evaluated in a number of malignancies as quantitative parameter to define diseases aggressiveness and its correlation with outcome in a number of cancers has been and is being investigated. The authors use a model in which genomic features are associated to radiomic features and clinical outcomes in patients with locally advanced SCC esophageal cancer treated with different neoadjuvant chemo and radiotherapy regimens.
The authors have a testing cohort, an internal testing set and an internal external set. Pateint population numbers are quite low. The paper describes a similar radiomic score in esophageal ACC described by Zhang et al 2020 used to predict lymph node metastasis on pretreatment PET (https://doi.org/10.1259/bjr20201042 ). This would be worth citing also because they use a radiomic score which has the same name Rad-score.
The paper could be of interests as it evaluates the prognostic ability of a mixed model to predict DFS/OS. Of interest and added value is the availability of pre and post neoadjuvant treatment CT scans If risk stratification
There are limitations based on the size of the sample tested. The title should be revised and contain what is in the discussion – pilot study.
The consecutio of the paper is not always clear; sentence structure should be revised overall. The limitation of a small sample size has to be clear in the paper, also because the authors cite as “small sample size (line 318) a study with 32 patients.
The article is complex and is not always written in a reader-friendly manner and confusing (see lines 104-108; 113-116; 307-308 as examples).
The section headers need to be clearer for ease.
1)Methods & Materials:
The section should contain only methods and materials and not results.
Please consider the use of “Radiomics” or “radiomic features” (not radiomics features) – the same is true for genomic features or genomics.
Patient selection: details on inclusion and exclusion criteria are needed.
Define the population Were these all consecutive patients with esophageal SCC presenting to the institution, then you excluded those who did not have CT prior to and post nCRT?
Starting line 121 – are these different schemes institutional standards of care? If yes, please explicit. If no, they are they selection criteria- detail
Line 130: this is a study not a “work”. Please give a number of IRB approval. Give statement regarding ethics, such as a reference to the Declaration of Helsinki.
Section 2.2 – “duration from randomization” line 139 – patients are randomized? To what?
Section 2.3 Radiomics features – would it not be best to call it Data extraction and then have the different parts – clinical parameters -radiomic features- genomic features
Add the features you have extracted in this sectiob
Which are the institutional differences you want to minimize referred to radiomics in line 151?
Sections 2.4 Resampling of images- line 153 – do not give the result –“small sample size” as is “distribution of DFS (7 out 21 having recurrence)” line 153 - reword if it a selection criterion
briefly describe the method used: you cannot explain the number of patients with DFS in the M&M unless this is exclusion criterion ad priori.
All images had CT scans had the same characteristics? Is there a limit of slice/pixel size for the images – voxel sizes?
Section 2.5 – move the statistics to the statistic section
Section 2.6 – we are introduced to a machine learning approach in the title , but this is feature selection-
Line 172 – “ in the training set” – the features found significant in the training set were used in the two test sets, correct? what would you like to say with machine learning approaches if you only did so in the training set
Describe your clinical model – you did not only use DFS and OS, you used your risk factors. You shoulddetail which parameters were collected in M&M.
Please revise tenses
Your final model is composed of radiomic feature model + genetic model + clinical model? The nomogram designs should be explained as not being “a nomogram” but nomograms were constructed
Sentence lines 190-192 should go in the statistics section as should the
Paragraph 2.3. Reconsider the wording in lines 144-147
2)Results
Define all colors codes in the figures. Data is presented in tabular manner and in figures. More details in the caption may be of aid.
The authors run three different testing sets with data from two institutions – the training set (n= 28 from Institution 1), the internal testing set (n=37 from Institution 1) and the external testing set (n= 41 from institution 2). The results are not reported appropriately and another evaluation may be necessary.
The training set may be compared to the internal testing set, but the internal testing set should be compared to the external testing set alone, and not to all patients from institution 1.
There are 2 chemotherapy options in both institutions –how could this have changed the response to treatment in terms of DFS/OS?
Did all patients finish their CRT? Detail timing of biopsy sample compared to start of treatment
Clinical risk factors: these are defined as? Those patient characteristics which are statistically significant? nodal involvement, T stage?
Section 3.3 Feature selection starting line 226: the training set had 37 patients and found “37 genes were differently expressed between patients with and without relapse…”? is this correct. At what time point was relapse evaluated and shown significant? Significance should be in parenthesis.
Section 3.4 – what if a classification model? Not described previously or not using the sane expression – state in the m&m that data will be evaluated with and without genetic features.
Suggestion to make a separate section for nomograms
Define Rad-score (radiomic score) – N staging is biopsy based – histological characteristic
Line 272 -define in methods and statistics that performance will be evaluated every six months, not in the results
Line 283: what are “diagnostic possibilities”? the likelihood of a correct diagnosis?
3)Discussion: need re-editing
The sample size is quite small, the validation set should be larger than the testing sets. The sets should be compared among them (testing vs. internal; testing vs. external; internal vs external) and not all from institution 1(testing and validation) vs institution 2 (validation) also in the table. There could be a number of other references added to the paper.
Risk stratification of the clinical nomogram is not significant in the training set but is in the other two sets – comment?
Line 376 – retrospective study form prospective clinical studies – this is not detailed in the methods.
Line 380: what are the “clinical variables” which are not balanced for the different institutions? Tehse should have been exclusion criteria – Table 1 says you have data for all of these characteristics.
Line 383 – please revise as this is not a demonstration – data you collected showed that genetic features can be used to correctly select radiomic features or associated with radiomics features to predict outcome.
Personalized prognostic ability should always be coupled with personalized treatment schemes based on genetic profiles.
Round 2
Reviewer 1 Report
Authors revise their paper along the lines reviewers commented. Their paper with the relatively small number of case is now ready for publication as a preliminary report.
Author Response
Thank you for the reviewer's comment. There is no specific reply comment. The paper has undergone minor editing including English proofreading/spelling check.
Reviewer 2 Report
The authors have tried to change the points risen. The inclusion and exclusion should be included in the main text and the figure for inclusion criteria cited (suppl). Please add specific references for each Supplementary table and or Figure.
"Randomization" referring to the cohort has to be deleted throughout the document (still described in line 159).
Line 147 - Maybe there answer should be added to the text or to supplementary material - which has been "widely implemented in genomic studies and has been recently shown to have an advantage in correcting for the difference in radiological feature values resulting from discrete image acquisition protocols across multicenter radiological studies [2]."
Author Response
Reviewer 2
The authors have tried to change the points risen. The inclusion and exclusion should be included in the main text and the figure for inclusion criteria cited (suppl).
Authors’ Reply: Thank you for the suggestion. We have revised accordingly in p.4, line 120-125.
Please add specific references for each Supplementary table and or Figure.
Authors’ Reply: Thank you for the suggestion. We have revised accordingly and made references at specific locations and also at the end of the manuscript (line 432-441).
"Randomization" referring to the cohort has to be deleted throughout the document (still described in line 159).
Authors’ Reply: Thank you for the suggestion. We have revised accordingly in line 164.
Line 147 - Maybe there answer should be added to the text or to supplementary material - which has been "widely implemented in genomic studies and has been recently shown to have an advantage in correcting for the difference in radiological feature values resulting from discrete image acquisition protocols across multicenter radiological studies [2]."
Authors’ Reply: Thank you for the suggestion. We have revised accordingly in line 154-157.